# 🐛 LLARVA: Vision-Action Instruction Tuning Enhances Robot Learning

**Dantong Niu**\*    **Yuvan Sharma**\*    **Giscard Biamby**    **Jerome Quenum**
**Yutong Bai**    **Baifeng Shi**    **Trevor Darrell** [†]    **Roei Herzig** [†]
Berkeley AI Research, UC Berkeley
Project Webpage: https://llarva24.github.io/

**Abstract:**

In recent years, instruction-tuned Large Multimodal Models (LMMs) have been successful at several tasks, including image captioning and visual question answering; yet leveraging these models remains an open question for robotics. Prior LMMs for robotics applications have been extensively trained on language and action data, but their ability to generalize in different settings has often been less than desired. To address this, we introduce LLARVA, a model trained with a novel instruction tuning method that leverages structured prompts to unify a range of robotic learning tasks, scenarios, and environments. Additionally, we show that predicting intermediate 2-D representations, which we refer to as *visual traces*, can help further align vision and action spaces for robot learning. We generate 8.5M image-visual trace pairs from the Open X-Embodiment dataset in order to pre-train our model, and we evaluate on 18 different tasks in the RLBench simulator as well as a physical Franka Emika Panda 7-DoF robot. Our experiments yield strong performance, demonstrating that LLARVA—using 2-D and language representations—performs well compared to several contemporary baselines, and can generalize across various robot environments and configurations.

**Keywords:** LMMs, Vision Action Instruction Tuning, Robot Learning

## 1    Introduction

Recently, instruction-tuned Large Multimodal Models (LMMs), such as InstructBLIP [1], Instruct-GPT [2], LLaVA [3, 4], PALM [5] and others have demonstrated state-of-the-art performance on a variety of vision-and-language tasks. However, existing LMMs for robotics [6, 7, 8, 9] do not always demonstrate the same success and consistency across various embodied settings. This may result from the unique challenges encountered in robotics, such as the variability of real-world environments, the differences between robots, and the need to control actions reliably. Since LMMs have been proven to be successful in part due to multimodal instruction tuning, it is natural to leverage this technique in a robotics setting as well. Here, we propose a vision-action instruction tuning method that can bridge the gap between a language model's fundamental pre-training objective—next-word prediction—and the goal of enabling the model to handle various robotics settings.

In this work, we introduce our *Large LAnguage model for Robotic Vision and Action* (LLARVA), an open-source instruction-tuned LMM for robotic applications that can generalize efficiently across various environments and robotic configurations. Our key idea is the formulation of a novel instruction prompt that encapsulates robot type, task, scene configuration, and control regime in a natural language prefix amenable to contemporary LMMs. We present an instruction tuning procedure tailored to the robotic domain: when given an instruction that describes the robot model, control mode, robot task, and proprioceptive information, the model needs to predict future actions given

---

\*Equal contribution. [†]Equal advising.

8th Conference on Robot Learning (CoRL 2024), Munich, Germany.

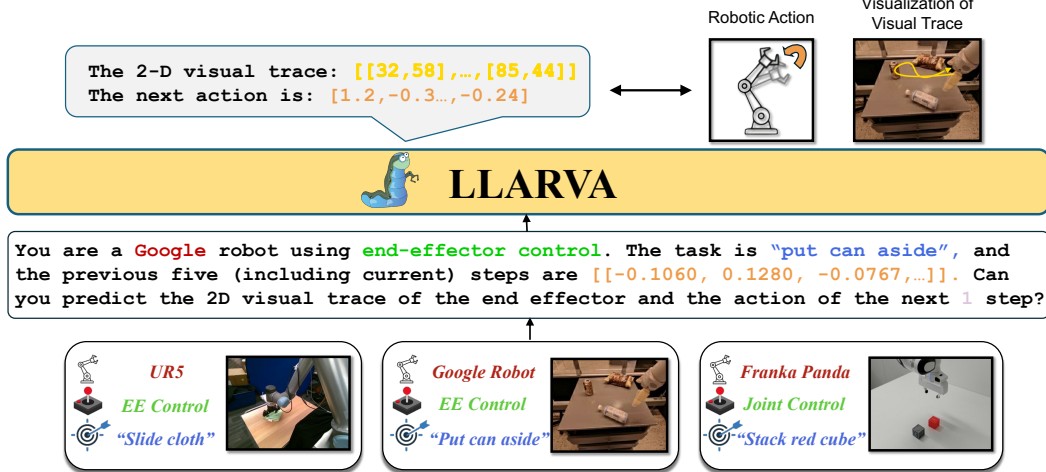

Figure 1: **Overview of LLARVA.** We introduce a novel instruction tuning method that leverages structured prompts to unify a range of robotic learning tasks, scenarios, and environments and 2-D visual traces to further align vision and action spaces. The model works via a language instruction that contains robot model, control mode, robot task, proprioceptive information, and number of predicted steps, and outputs text with the next robot action(s) and the visual trace for the remainder of the episode.

the natural language prompt. This architecture allows us to leverage structured language prompts as a "lingua franca" for robotic perception and control (See Figure 1).

However, aligning the vision and action modalities to produce meaningful robotic outputs is still not a trivial task. Though recent robotic models have used 3-D representations such as voxels and point clouds to overcome this, these representations are difficult to incorporate into most existing open-source LMMs because they typically accept a single image plus language as input. For these reasons, we utilize 2-D images, which are easy to scale and integrate with existing LMMs.

We find that predicting an intermediate 2-D representation, which we refer to as visual traces, can help align the vision and action spaces across different robot and task configurations. In particular, we generate the 2-D visual trace (projection) of an end-effector and force the model to predict this trace alongside the next robot action(s). This waypoint prediction helps align each robotic action to the end-effector's location, allowing the model to focus on fine-grained localization and resulting in a more accurate prediction of robot actions. To achieve this, we construct instructions with such visual traces using the Open X-Embodiment dataset (OXE) [10], carefully categorizing the action space, the robot type, and the control type.

Through empirical study, we show that our vision-action instruction tuning approach using structured prompts leads to generalization across various robot environments and configurations. Additionally, we show that predicting visual traces can help further align vision and action spaces. We evaluate LLARVA on 18 different tasks in RLBench's simulated environment and on picking, stacking, and destacking tasks with a real 7-DoF Franka Emika Panda robot. Finally, we evaluate our model's generalization across two robots in RLBench on four tasks. We show that LLARVA—using 2-D and language representations—performs well compared to several contemporary baselines.

## 2 Vision-Action Instruction Tuning

### 2.1 Preliminaries

LMMs are designed to handle multiple data modalities simultaneously, such as images and their corresponding text descriptions. Each modality is encoded into a shared embedding space, which is then utilized for reasoning by a language model $f$ parameterized by $\theta$. Specifically, an image is encoded using a pre-trained visual encoder, denoted as $v$ parameterized by $\phi$. A corresponding text

description is tokenized and encoded using a fixed language encoder $e$ parameterized by $\gamma$. Given an input image $o$ and a language task description $l$, the language model generates a text response $R$ as follows: $R = f_\theta(v_\phi(o), e_\gamma(l))$.

In this paper, we use an LMM within the context of robotic episodes, which are characterized by temporal sequences of visual observations $o_{1:N}$ and proprioceptive states $s_{1:N}$. Here, $N$ denotes the length of an episode. Notably, in the realm of LMMs for robotics applications, the output $R$ typically encompasses one or more predicted actions for an episode. Next, we describe our LLARVA model.

## 2.2 LLARVA Model

**2-D Visual Traces**. Visual traces play a key aspect in our vision-action instruction methodology. The choice of 2-D traces is made to match the high availability of image-based large robotics datasets such as OXE, but our method can also be implemented with 3-D data. To achieve alignment between visual inputs and robotic actions, we predict visual traces as an auxiliary task, as we find that this helps to gain better fine-grained localization, resulting in a more accurate prediction of robot actions.

We define *2-D Visual Traces* as a sequence of coordinates $(x, y)$ in a two-dimensional space, which is aligned with the input image $o_t$ at time step $t$. These coordinates represent the trajectory of the gripper (or end-effector, hand, etc.) throughout the episode. The visual trace at timestep $t$ is:

$$\mathcal{P}_{t:N} = \{(x_i, y_i) \mid i = t, t+1, \ldots, N\} \tag{1}$$

Here, $(x_i, y_i)$ denotes the $i-th$ coordinate in the entire visual trace for the episode, and $N$ represents the number of time steps in the episode. We note that language model decoders are crucial in converting multimodal inputs into actionable outputs in robotics. By leveraging the shared vision-action embedding space, our decoder produces responses that the robotic system can use.

**Input**. The input to our LLARVA architecture comprises two components. First, we have the visual observation $o_t$, an image capturing the state of the environment at timestep $t$. Second, we have the language instruction input $l_t$, which prompts the model to forecast a specified number of subsequent steps, integrating embodied information such as the robot, control mode, and previous proprioceptive states as well as the task directives. Specifically, we formulate an instruction template featuring the robot type $\mathcal{R}$ (e.g., Franka, UR5, xArm), control mode $\mathcal{M}$ (e.g., joint or end-effector control, absolute or delta control), task instruction $\mathcal{I}$ (e.g., "open the drawer"), proprioceptive information $\mathcal{S}$ (e.g., positions or velocities), and a query indicating the number of future actions to predict, denoted as $n$. The complete instruction is formulated as follows:

> $l_t$ = "You are a $[\mathcal{R}]$ robot using $[\mathcal{M}]$ control. The task is $[\mathcal{I}]$, and the previous $[h]$ steps are $[\mathcal{S}]$. Can you predict the trajectory of the end-effector and the action of the next $[n]$ steps?"

To develop a versatile and adaptive framework capable of accommodating training for tasks with varying time horizons, we add flexibility to the proprioceptive information input. Specifically, this information is structured as $\mathcal{S} = s_{t-h:t}$, representing a sequence of past joint and/or gripper states. Here, $h$ is the number of previous time steps the model is conditioned on, and is decided based on the task. This approach ensures robustness and adaptability across a spectrum of task durations, enabling effective training for both short-term and long-term objectives.

**Architecture**. Our objective is to develop a model capable of predicting robotic actions that exhibit generalization across a diversity of robotic tasks, scenarios, and environments. The model architecture is illustrated in Figure 2. Our instruction-tuned model $\pi$ is designed to leverage both the current visual observation $o_t$ and the accompanying language instruction $l_t$ as input. Subsequently, it predicts the action sequence for the next $n$ steps $\mathcal{A}_{t:t+n-1}$ and the future 2-D Visual Traces of the end-effector $\mathcal{P}_{t:N}$, spanning from the current step to the final step within the episode:

$$\pi(o_t, l_t) \rightarrow \mathcal{A}_{t:t+n-1}, \mathcal{P}_{t:N} \tag{2}$$

where $l_t$ is constructed as defined above.

In our proposed pipeline, the input image undergoes processing by the frozen vision encoder $v_\phi(\cdot)$, which extracts visual features and projects into a latent space via an MLP layer $\mathcal{H}$. This aligns the visual features with the dimensionality of the language tokens. Simultaneously, the language input undergoes tokenization using a language encoder. The visual tokens and word tokens are then concatenated and fed into the auto-regressive transformers of the LMM $f_\theta$, which are trained for next-token prediction.

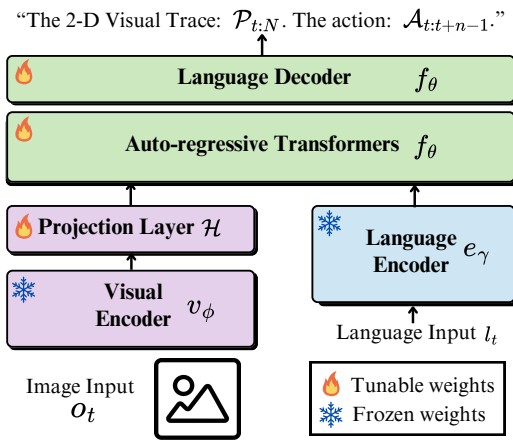

Figure 2: **Architecture of LLARVA.**

## 2.3 Training

While keeping the vision encoder and language encoder frozen, we use instruction tuning to train the auto-regressive transformers using standard LoRA adapters [11] for both the pre-training and fine-tuning stages. Each image $o_t$ for an episode is accompanied by a language instruction $l_t$, and the predictions consist of robotic actions $A_{t:t+n-1}$ and visual traces $\mathcal{P}_{t:N}$. Next, given $o_t$ and $l_t$, we predict the next actions and 2-D visual trace. Specifically, for a response $R$, we compute the probability of the target actions and target visual traces by the following equation:

$$p(A_{t:t+n-1}, \mathcal{P}_{t:N} \mid o_t, l_t) = \prod_{i=1}^{|R|} p_\theta(x_i \mid o_t, l_t) \tag{3}$$

where $\theta$ represents the trainable parameters, $x_i$ is the current prediction token, and $n \leq N$. To calculate loss, we use the standard cross-entropy function with these probabilities. Next, we describe our two-step training process, the large-scale pre-training and the fine-tuning for a downstream task.

**Step 1: Vision-Action Instruction Pre-training.** We begin with an LMM that has been pre-trained on vision-language (VL) tasks. In order to generalize across robotic tasks, scenarios, and environments, the model is pre-trained on our large-scale vision-action instruction dataset. Due to the diversity of this dataset, our model is trained simultaneously for multiple configurations of prompt variables such as robot type $\mathcal{R}$, control mode $\mathcal{M}$ or task instruction $\mathcal{I}$ [1]. Using language as input allows us to bridge fundamental gaps between subsets brought by these different configurations. This extensive and varied training process can establish a powerful LMM framework that can be further fine-tuned and adapted to handle various robotic settings. We note that this pre-training stage is different from standard LMM pre-training. As opposed to aligning the two modalities using a projector in VL, here we align the two modalities for generalizing robotic configurations.

**Step 2: Fine-tuning for Downstream Tasks.** Unlike other fields, a robotic model must be fine-tuned on a downstream task before it can be evaluated due to the practical considerations of real-world physical properties. Therefore, we fine-tune the pre-trained model using a small dataset with a fixed configuration for the factors defined in Section 2.2 (e.g., the instruction has the same robot type $\mathcal{R}$, control mode $\mathcal{M}$, etc.). Having seen diverse data samples makes it easy for the model to adapt to specific downstream settings resembling what it has already encountered in pre-training. Given new tasks, environments, or robot types, LLARVA can be adapted by fine-tuning on some example demonstrations. In addition, data from new modalities such as 3-D, depth or tactile information can potentially be incorporated and utilized in the fine-tuning by modifying the instruction template.

When LLARVA is to be used for a new setup or environment, this fine-tuning must first be carried out using example demonstrations.

---

[1]Details of the configuration for each subset are available in Section B of the Supplementary Material.

| Method | Task | | | | | | | | | | | |
|---|---|---|---|---|---|---|---|---|---|---|---|---|
| | open drawer | meat off grill | turn tap | put money | push buttons | sweep dustpan | slide block | close jar | screw bulb | place wine | reach and drag | stack blocks |
| *3-D methods* | | | | | | | | | | | | |
| C2FARM-BC | 20 | 20 | 68 | 12 | 72 | 0 | 16 | 24 | 8 | 18 | 24 | 0 |
| PERACT | 80 | 84 | 80 | 44 | 48 | 56 | 72 | 60 | 24 | 12 | 68 | 36 |
| *2-D methods* | | | | | | | | | | | | |
| Image-BC (CNN) | 4 | 0 | 8 | 4 | 0 | 0 | 4 | 0 | 8 | 4 | 0 | 0 |
| Image-BC (ViT) | 0 | 0 | 16 | 0 | 0 | 0 | 0 | 0 | 16 | 0 | 0 | 0 |
| LLARVA | **60** | **80** | **56** | **44** | **56** | **84** | **100** | **28** | 8 | 12 | **52** | **0** |

Table 1: **Success rate (%) on RLBench Multi-Task setting.** We fine-tuned (with visual trace prediction) the pre-trained model on 12 tasks and evaluate with 25 episodes per task. Each evaluation episode is scored either 0 for failure or 100 for success. We gray out methods with 3-D information.

## 2.4 Vision-Action Instruction Dataset

In order to pre-train LLARVA, we generate 8.5M image-visual trace pairs from the Open X-Embodiment (OXE) dataset [10]. As shown in Figure 6 in Supplementary, our dataset consists of images from a diverse collection of 37 OXE subsets with 13 different robots, including a wide assortment of tasks, environments, cameras (and thus images), and end-effectors, among other factors. For each image in an episode, we calculate the 2-D visual trace of the end-effector $\mathcal{P}_{t:N}$. For this purpose, we use a bounding box detector [12] that is trained specifically on each of the different end-effectors in OXE. The center points of bounding boxes are used for a simpler representation, and the visual trace for step $t$ is then the ordered list of all center points from image $t$ to image $N$. More details of the dataset are shown in Section B in Supplementary.

# 3 Experiments and Results

We evaluate LLARVA on 18 tasks in RLBench and compare to both existing 2-D and 3-D models. In addition, we also test on a real 7-DoF Franka Emika Panda robot.

## 3.1 Implementation Details

LLARVA is implemented using PyTorch [13] with the official LLaVA 1.5 [4] implementation. The base LMM uses a Llama 2 7B-parameter LLM, the default image projection layer, and the CLIP ViT-L/14 vision encoder. We use 8 NVIDIA A6000 GPUs for training, and 1 A6000 GPU for evaluation. Additional information, such as training and fine-tuning recipes, are in Supplementary Section C.

## 3.2 RLBench Evaluation

**Experimental Setup**. We evaluate using the same 18 RLBench tasks as in [7]. In the fine-tuning stage, the front view is chosen as $o_t$, conditioned on the previous 5 joint positions (*i.e.* $h = 5$). LLARVA predicts the visual trace and the next action step (*i.e.* $n = 1$), which is an 8-dimensional vector consisting of 7 joint velocities and a binary gripper state. For evaluation, we take 25 episodes per task in the validation set and score each episode either 0 for failure or 100 for success. We use 5 seeds, which are averaged to get the final success rate.

**Baselines**. We compare LLARVA to several baselines using 2-D and 3-D information. Image-BC (CNN) and Image-BC (ViT) [14] are 2-D language-conditioned models that use CNN and ViT vision encoders, respectively, reported in PerAct [7]. The PerAct [7] model uses voxels as 3-D input to calculate actions, as does C2FARM-BC [15]. In contrast to the 3-D line of work, our input uses only one camera view without any 3-D information.

**Results**. We show results for 12 of the tasks in Table 1; additional long-horizon task results (such as those with multiple steps or subtasks) are presented in appendix A. Our model completely out-

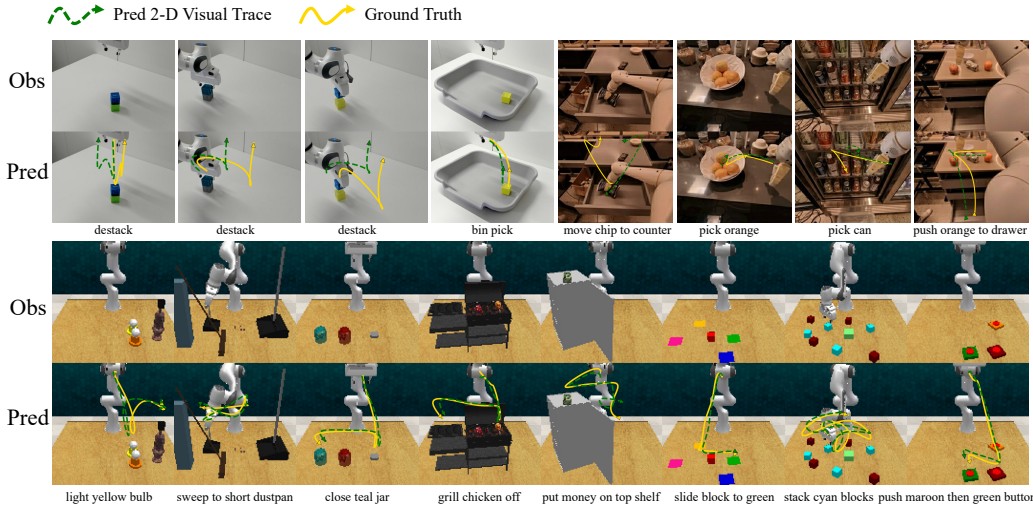

Figure 3: We visualize the ground truth (yellow line), and **predicted 2-D visual trace (green dash line)** after downstream tasks fine-tuning. Our predicted 2-D visual trace gives reasonable route planning to achieve the goal, even sometimes diverges from the ground truth.

| Method | 2-D Visual Trace | Task | | | | | |
|---|---|---|---|---|---|---|---|
| | | pick | stack | destack | one button | two buttons | three buttons |
| RPT [16] | - | 87.50 | 31.25 | 93.75 | - | - | - |
| Octo [8] | - | 56.25 | 12.5 | 37.5 | 53.75 | 8.75 | 0 |
| LLARVA | ✗ | 81.25 | 50.00 | 87.50 | 96.25 | 68.75 | 32.5 |
| LLARVA | ✔ | **93.75** | **56.25** | **100** | **97.5** | **83.75** | **58.75** |

Table 2: **Success rate (%) of LLARVA on a real robot.** We compare LLARVA with RPT and Octo by taking each pre-trained model and fine-tuning them on the same set of demonstrations. LLARVA outperforms the others on all the tasks.

performs other 2-D based methods: we achieve an average success rate of 43.3%, while Image-BC (CNN) and Image-BC (ViT) both achieve 1.3%. In addition, LLARVA competes with and even beats 3-D based methods, like C2FARM-BC, which averages 22.7%, and PerAct, which achieves 55.3%. We note that 2-D methods here also include those that use multiple input images from different camera views, while LLARVA uses only one camera view and displays stronger performance. Moreover, we observe that despite some cases of occlusion that arise from using just one camera view, LLARVA can still succeed in such situations, showing the adaptability of our model.

### 3.3 Real Robot Evaluation

**Experimental Setup**. Following the setting in [16], we use a 7-DoF Franka Emika Panda robot with the default 1-DoF parallel jaw gripper. The input image $o_t$ comes from the right side RGB camera. We evaluate LLARVA on three pick tasks: "pick cube", "destack cube", and "stack cubes", and another task "push buttons". During fine-tuning, we use the same number of episodes (1920) as [16] for pick tasks. The "push buttons" task is a more complicated real world task than the pick tasks as it requires the model to push up to three buttons from four color variations (red, blue, green, yellow) in a specified sequence. The setup is shown in Figure 4 in Supplementary. In our experiment, we use our pre-trained weights on OXE to fine-tune the model with a total of 450 episodes (150 episodes each for the one, two, and three button tasks).

During training, we condition on joint positions of the previous 16 steps, and predict 7-dimensional delta joint positions and 1-dimensional gripper status for the following 16 steps. To calculate the final success rate, we evaluate LLARVA over 16 episodes and take the average across 5 runs.

| Instruction Pre-Training | 2-D Visual Trace | Task | | | |
|---|---|---|---|---|---|
| | | reach and drag | place wine | meat off grill | slide block |
| ✘ | ✘ | 40 | 8 | 36 | 48 |
| | ✔ | 48 | 12 | 40 | 76 |
| ✔ | ✘ | 44 | 4 | 56 | 80 |
| | ✔ | **52** | **12** | **80** | **100** |

| Robot | Task | | | |
|---|---|---|---|---|
| | sweep dustpan | put money | push buttons | meat off grill |
| Franka | 84 | 44 | 56 | 80 |
| Sawyer | 80 | 48 | 48 | 72 |

Table 3: **Left: The effect of the instruction pre-training and 2-D visual trace.** We ablate the instruction pre-training (step 1 in Section 2.3) and the visual traces for four random tasks in RLBench. **Right: Evaluation across different robots.** We show the success rate (%) of four randomly chosen tasks in RLBench. Using the same training recipes, we fine-tune our pre-trained model with visual trace but different robots. For evaluation, we take 25 episodes per task, and each of them is scored as 0 for failure or 100 for success.

**Baselines**. We compare LLARVA with RPT [16] and Octo [8], both of which claim the benefit that pre-training brings to the downstream tasks. RPT uses a BERT-like [17] formulation and must additionally pre-train on in-domain data, while Octo is pre-trained on the mixed dataset OXE and maps the various configurations to the same action space. In contrast, LLARVA is pre-trained in a more diverse manner, without imposing a single action space, via a unified language template.

**Results**. We show the results in Table 2. LLARVA achieves the highest success rate across all pick tasks. Additionally, compared to RPT, LLARVA only needs one unified pre-trained model that is first instruction-tuned on our dataset and then fine-tuned on each downstream task. In contrast, RPT needs to be separately pre-trained on each individual task before being adapted to them, which brings additional time and computation cost. In addition, Octo has difficulties adapting to a downstream task using a different control mode as the model has only seen end-effector control in the pre-training stage. LLARVA shows better quality both in terms of efficiency for adapting to downstream tasks and compatibility/generalization with different control modes.

**Robot Planning with 2-D Visual Traces**. To align the vision and action spaces, we predict the end-effector's visual traces, which forces the model to develop a more comprehensive understanding. In Figure 3, we show through visual traces that our model can plan alternative yet correct paths. For example, in the top left image, the gripper takes a different path (go left) than the ground truth (go right), yet it still succeeds in destacking the cube. Figure 3 also shows qualitatively that the visual trace can help with long-horizon tasks by acting as a memory buffer that compensates for the limited number of previous robotic states the model can handle. For instance, in the bottom right image in Figure 3, the task is "push maroon button, then push green button." It can be seen that the visual trace helps the model reach the "green button" after finishing the first subtask.

### 3.4 Ablations

**The Effect of Instruction Pre-training**. We ablate the instruction pre-training in Section 2.3 to evaluate its effect. As shown in Table 3, our model can achieve an *average improvement of 17.5% across the four tasks* regardless of visual traces. We believe this improvement is due to our diverse, large-scale pre-training, which leverages language instructions to create a strong robotics LMM backbone that can successfully adapt to downstream robotic settings.

**The Effect of 2-D Visual Traces**. To test the importance of visual traces, we fine-tuned the model with and without the visual trace prediction task. Table 3 shows that the traces provide a 15% average improvement across the four tasks. Complete visual trace ablation results with additional analysis can be found in Table 5 and appendix A.1. In particular, an improvement can be seen in tasks with frequent occlusion of objects such as "meat off grill" and "slide block." We note that using both pre-training and 2-D visual traces can help the model complete the task even when the target object is not visible. The "put money on top shelf" example in Figure 3 demonstrates this: the shelf is not visible, and the gripper becomes occluded once it moves behind the counter. While the

results show that visual traces increase performance across all tasks tested, we find that the greatest benefit of visual traces occurs in "meat off grill" and "slide block", both tasks with distractor objects.

**Cross-Robot Generalization**. To evaluate LLARVA's generalization across different robots, we randomly selected four tasks in RLBench, and finetuned and evaluated on *Sawyer* instead of *Franka Emika Panda*. As shown in Table 3, LLARVA achieves similar results across both robots, which further proves the generalization brought by our large-scale vision-action instruction pre-training.

## 4 Related Work

**Instruction Tuning**. LLMs [18, 19, 20] are typically pre-trained on a large corpus of text with an unsupervised training objective of next-word prediction. Instruction tuning (IT) helps to bridge the gap between the language model's fundamental pre-training objective of next-word prediction and the user's goal of having the model perform specific tasks using input-output pairs whose inputs include text phrased as instructions. Flamingo [19], GPT-4 [21], and BLIP [22, 23] were pioneering early LMMs, and LLAVA [24] prompted GPT-4 with image-caption pairs to generate multimodal instruction tuning data. Many recent VL models [1, 25, 26, 27, 28, 29] have followed this approach, as IT has been shown to improve generalization in zero-shot and few-shot tasks [2, 30, 31]. Unlike these works, here we construct instructions from a custom template that we fill in with relevant information about the robotic episode.

**Language-conditioned Robot Agents**. Recent language-conditioned models for robotics have tried different approaches, such as generative video pre-training (GR-1 [32]), tokenized voxels for incorporating 3-D information (PerAct [7]), co-training on internet-scale VL data (RT-2 [9]), and large-scale diverse pre-training with a transformer architecture (Octo [8]). However, each approach has distinct drawbacks that we attempt to address in our model. For instance, GR-1 requires architectural changes to account for different action spaces when fine-tuning, while LLARVA is more flexible and does not need such changes. Moreover, PerAct relies on 3-D information unavailable on a large scale across environments, which is why we chose to use 2-D images when implementing our model. RT-2 uses co-training on a separate web-based dataset, while we more efficiently use instruction-tuning on a smaller, diverse vision-action instruction dataset to create a strong backbone. Lastly, Octo uses a specialized head to predict actions, while the language head in our model is trained to predict both robotic actions and visual traces in "natural" language.

**Trajectory Prediction using Object Representations**. Trajectory modeling has been a fundamental aspect of many machine vision applications. Many previous works have investigated the use of object representations for trajectory modeling, ranging from classical image understanding tasks [33, 34, 35] to an object-centric approach for video understanding [36, 37, 38, 39, 40] with object tracking and interactions to even scene graphs [41, 42, 43, 44]. More recently, there has been an increased interest in trajectory modeling effectiveness for vision and language, such as the referring expression localization task [45, 46, 47] and semantic segmentation using text prompts [48, 49]. In particular, the localized narrative dataset [50] enabled a new task that aims to align a long and detailed image caption to a human trajectory. Our work takes a different approach as we use LMMs to exploit the concept of a 2-D visual trace for implicit end-effector object representation.

## 5 Conclusion

Our proposed model, LLARVA, represents a significant advancement in the application of instruction-tuned LMMs for robotics. By leveraging structured prompts to unify a range of robotic configurations and introducing the concept of visual traces, we have demonstrated a generalization method that better aligns vision and action modalities. Our extensive training on 8.5M image-visual trace pairs derived from the Open X-Embodiment dataset and evaluation in both simulated and real-world settings highlight the model's superior performance and generalization capabilities compared to existing approaches. This work marks a meaningful step forward in the integration of LMMs with robotics, promising enhanced adaptability and efficiency in various robotic applications.

**Acknowledgments**

We would like to thank Max Fu for his assistance with data collection, Ritwik Gupta for his compute support, as well as Xiaolong Wang and Ilija Radosavovic for their helpful feedback and discussions. We also thank Nicole Walters for creating our lovely logo.

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

# Supplementary Material for LLARVA

Here, we provide additional information about our experiments, our model's emergent properties, our constructed dataset, and implementation details. Specifically, Section A provides additional experiment results, Section B provides details about our constructed vision-action instruction dataset, and Section C provides additional implementation details.

## A  Additional Experiment Results

### A.1  Additional Experiments

| Method | | | | | Task | | | | |
|--------|------|----------|------|-------|-------|-------|-------|-------|-------|
| | open drawer | meat off grill | turn tap | put money | push buttons | sweep dustpan | slide block | close jar | screw bulb |
| *3-D methods* | | | | | | | | | |
| C2FARM-BC | 20 | 20 | 68 | 12 | 72 | 0 | 16 | 24 | 8 |
| PERACT | 80 | 84 | 80 | 44 | 48 | 56 | 72 | 60 | 24 |
| *2-D methods* | | | | | | | | | |
| Image-BC (CNN) | 4 | 0 | 8 | 4 | 0 | 0 | 4 | 0 | 8 |
| Image-BC (ViT) | 0 | 0 | 16 | 0 | 0 | 0 | 0 | 0 | 16 |
| LLARVA | **60** | **80** | **56** | **44** | **56** | **84** | **100** | **28** | **8** |

| Method | | | | | Task | | | | |
|--------|------|----------|------|-------|-------|-------|-------|-------|-------|
| | place wine | reach and drag | stack blocks | put in drawer | sort shape | insert peg | stack cups | put in cupboard | place cups |
| *3-D methods* | | | | | | | | | |
| C2FARM-BC | 8 | 24 | 0 | 4 | 8 | 4 | 0 | 0 | 0 |
| PERACT | 12 | 68 | 36 | 68 | 20 | 0 | 0 | 16 | 0 |
| *2-D methods* | | | | | | | | | |
| Image-BC (CNN) | 0 | 0 | 0 | **8** | 0 | 0 | 0 | 0 | 0 |
| Image-BC (ViT) | 0 | 0 | 0 | 0 | 0 | 0 | 0 | 0 | 0 |
| LLARVA | **12** | **52** | **0** | **0** | **0** | **0** | **0** | **0** | **0** |

Table 4: **Success rate (%) on RLBench Multi-Task setting.** We fine-tuned (with visual trace prediction) the pre-trained model on 18 tasks and evaluate with 25 episodes per task. Each evaluation episode is scored either 0 for failure or 100 for success. We gray out methods with 3-D information.

**Complete Simulation Results on RLBench**. Following [7, 15], we evaluate LLARVA on 18 tasks in RLBench, with the comprehensive results presented in Table 4. LLARVA demonstrates significant improvements over 2-D methods and shows comparable performance to 3-D methods in most tasks. However, for "long horizon" tasks, which are more complex and involve multiple sub-steps or extended durations, LLARVA exhibits similar limitations to other methods. Specifically, for the "place cups" task, where success is defined by placing a specified number of cups on a rack, our experiments reveal that the model often successfully places the first cup but then becomes confused and wanders randomly. As discussed in the main paper, while the introduction of 2-D visual traces provides the model with a rudimentary sense of memory (as seen in the "push buttons" case), the length of this memory remains limited along the temporal axis. This issue can be partially addressed by incorporating additional information from previous steps; however, it will place greater demands on the maximum context length that vision-language models (VLMs) can handle. Fortunately, several recent works, such as [51], have demonstrated promising solutions and performance for VLMs with long context length.

**Real Tasks: Push Buttons**. We evaluate LLARVA on "push buttons", a more complicated real world task than the pick and place tasks involving cubes. This task requires the model to push up to three buttons from four color variations (red, blue, green, yellow) in a specified sequence. The setup is shown in Figure 4. The variation in the number, order and color of buttons makes the task

as well as the required motion of the end-effector more complicated. In our experiment, we use our pre-trained weights on OXE to fine-tune the model with a total of 450 episodes (150 episodes each for the one, two, and three button tasks). In our evaluation, we have three tasks: push one button given a specific color, push two buttons given specific colors and order, and push three buttons given specific colors and order. Results are presented in Table 2.

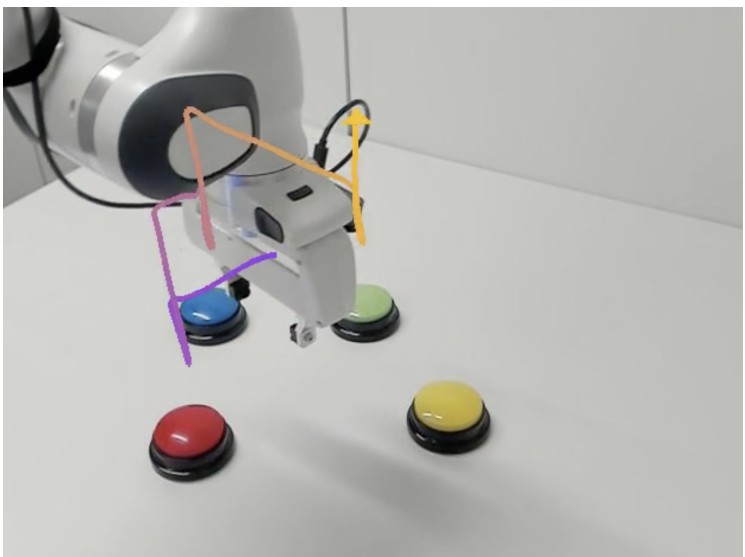

Figure 4: LLARVA completing the task "push red button, then blue button, then green button" on the Franka 7 DoF setup. The predicted 2-D visual trace is also visualized, with purple representing the current timestep and yellow representing the predicted end of episode.

**Visual Trace Ablations for 12 RLBench Tasks.** For a more comprehensive ablation study on the 2-D visual traces, we extended our ablation study described in Section 3.4 to 12 tasks. We present these results in Table 5. Overall, we find an average improvement of 13.7% with the visual traces, which further demonstrates their utility. We further note the improvement in tasks with distractor objects/variations such as meat off grill, open drawer, and slide block, and long horizon tasks such as push buttons.

| 2-D Visual Traces | open drawer | meat off grill | turn tap | put money | push buttons | sweep dustpan | slide block | close jar | screw bulb | place wine | reach and drag | stack blocks |
|---|---|---|---|---|---|---|---|---|---|---|---|---|
| ✘ | 36 | 80 | 72 | 56 | 52 | 24 | 44 | 0 | 24 | 4 | 24 | 0 |
| ✔ | 60 | 100 | 84 | 80 | 56 | 28 | 52 | 8 | 44 | 12 | 56 | 0 |

Table 5: Results for the visual trace ablation study across 12 tasks in the RLBench setting.

**Vision and Language Modality Ablations.** We performed an ablation study on the vision and language modalities to test the importance of each. For the vision-only ablation study, when fine-tuning, we remove the language instruction mentioned in the main paper and only feed in the visual input (images) to the model; the ground truth output stays unchanged. We evaluated on the simulated 12 RLBench tasks, and this model has a 0% success rate. The main reasons for this are: (1) the model does not receive any language instruction, and therefore cannot determine what task to perform, and (2) the model is no longer guided by the proprioceptive information acquired during the previous steps. We also evaluated the model on the real Franka Panda 7 DoF setup for the three tasks, and it performed poorly in this case as well. Results are presented in Table 6. It can be seen that tasks that have several variations (such as several cubes) are very hard to complete.

For the language-only ablation study, we do not input any images to the model; we only feed the language instructions. This model again has a 0% success rate across both RLBench and real tasks.

We note that this result is expected since the model is unable to see the environment, and therefore it can never know where exactly the robot is with respect to the target objects.

| Model | pick | stack | destack |
|---|---|---|---|
| LLARVA (language-only) | 0 | 0 | 0 |
| LLARVA (vision-only) | 5 | 0 | 3.75 |
| **LLARVA (vision & language)** | **93.75** | **56.25** | **100** |

Table 6: Results on vision and language ablation studies conducted on our real world Franka setup.

**Additional Explorations for Behavior-specific Tasks**. In order to provide a more comprehensive evaluation of LLARVA, we explored additional tasks in RLBench with specific behavior patterns. The results for 5 additional tasks are presented in Table 7, using the same fine-tuning and evaluation settings as in the main paper. These 5 tasks are categorized into two types: "Bending Tasks" and "Placement Tasks". In "Bending Tasks," the robot arm is required to grab the target object and move it down to a certain height. LLARVA demonstrates excellent performance on these tasks. In "Placement Tasks," the robot arm must grab the target object and move it to a pre-specified area. LLARVA performs well overall, except in cases requiring delicate operations during either the "grab" or "place" stages. For example, in the "put knife" task, most failures occur because the gripper misses the thin and delicate handle of the knife. Conversely, in the "put umbrella" task, most failures occur during the "place" stage, as the umbrella stand has a very small hole requiring precise positioning of the gripper during insertion. These issues are primarily due to the lack of detailed information from the visual observation, given that LLARVA uses only a single view image with a 128x128 resolution. Our future work will try to enable our VLM to process multiple view images or to adapt it with a more informative vision encoder that can better capture task-related features.

| 2-D visual Trace | Bending Task | | Placement Task | | |
|---|---|---|---|---|---|
| | toilet seat down | close laptop lid | put knife | put umbrella | move hanger |
| ✘ | 88 | 56 | 36 | 0 | 88 |
| ✔ | 96 | 68 | 40 | 4 | 88 |

Table 7: **Evaluation results on more tasks in RLBench.** We explore additional tasks in RLBench, which can be further categorized into "Bending Task" (*i.e.* the robot arm is supposed to grab the target object then bend and move the target down) and "Placement Task" (*i.e.* the robot arm is supposed to grab the target object, hold and move it to a specified area).

## A.2 Emergent Properties

**Multiple Attempts after Failure**. The top row of Figure 5 shows an example of the model failing to pick up an object, and then retrying as soon as the end-effector comes back into view without the object in its grasp. Specifically, the third image represents the moment where the end-effector becomes visible again, and LLARVA then attempts to complete the task again. This behavior emerges from the fact that the instruction prompt fed into LLARVA at this moment is similar to the prompt at the start of the first attempt, with the main difference being the previous actions/positions included in the prompt. We highlight this as an interesting emergent property since the training data does not include any examples with such behavior.

**Handling Obstructed Views**. In both pre-training and fine-tuning stages of LLARVA, we use only a single camera view to provide visual inputs. Using only 2-D inputs creates a challenge since robotics tasks require very accurate action predictions in three dimensions.

The model should be able to see the exact location of the target and also have a sense of depth, which other works typically achieve by using either multiple camera views or 3-D representations.

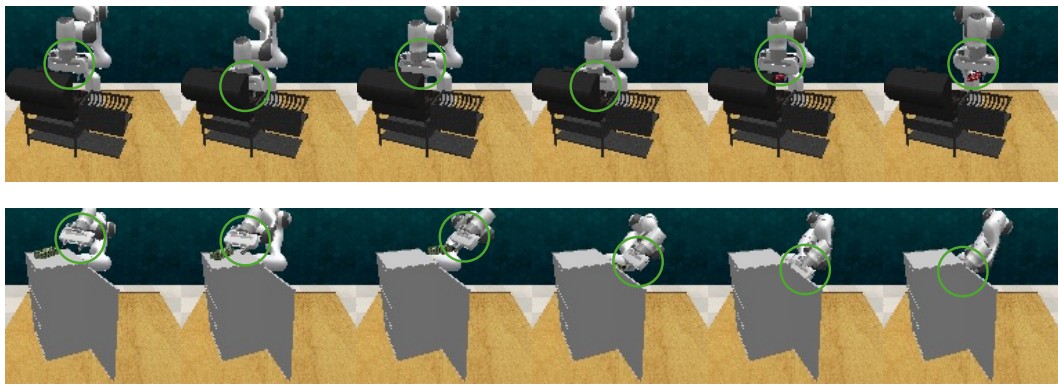

Figure 5: **Emergent Properties. Top**: The task in this episode is "take the steak off the grill". As we see in the third image, LLARVA fails to pick up the steak in the first attempt. However, it tries again and succeeds the second time, showing capability of attempting a task multiple times. **Bottom**: The single view which is used as the model input shows the top and back of a safe. The task is for the robot to move the money stack from the top of the safe to one of the shelves inside the safe. LLARVA can still move the money to the correct shelf in the safe despite the camera view not showing the shelves.

We note that our model uses a single camera view due to input limitations of current open-source LMMs. These limitations can certainly be overcome and are left for future work.

Using a single camera view presents further challenges when objects in the scene occlude each other. However, we find that LLARVA can often complete tasks even in these occluded situations. For example, as shown in the second row of Figure 5, the task is *"put the money away in the safe on the top shelf"*. The camera view only shows the top and back side of the safe, which is enough information to pick up the money from the top of the safe. However, the top shelf of the safe is not visible, and LLARVA can still predict the correct actions to place the stack of money there. This example shows LLARVA can, in some cases, work despite visual obstructions, which we believe is in part attributable to the introduction of the visual traces. We hypothesize that understanding and predicting the visual trace provides the model with information about a successful end-effector trajectory in the presence of such occlusions.

## A.3 Efficiency Analysis

| Step | Vis. Trace | Norm.Single Prompt Time | Norm.No.of Prompts/Episodes | Norm.Total Inference Time | Succ. Rate |
|------|------------|-------------------------|-----------------------------|---------------------------|------------|
| 4    | ✗          | 0.16                    | 4                           | 0.64                      | 87.5       |
|      | ✔          | 0.50                    | 4                           | 2.01                      | 97.5       |
| 8    | ✗          | 0.29                    | 3                           | 0.86                      | 85         |
|      | ✔          | 0.63                    | 3                           | 1.89                      | 90         |
| 12   | ✗          | 0.53                    | 2                           | 1.06                      | 82.5       |
|      | ✔          | 0.83                    | 2                           | 1.67                      | 92.5       |
| 16   | ✗          | 0.63                    | 1                           | 0.63                      | 81.25      |
|      | ✔          | 1                       | 1                           | 1                         | 93.75      |

Table 8: Efficiency analysis results for LLARVA with varying number of predicted steps. (*Norm. denotes Normalized, No. denotes the number of.)

In order to explore the inference efficiency with respect to the number of predicted steps, we conduct experiments on our real Franka setup with the pick cube task. The results are shown in Table 8. We vary the number of predicted steps with n = 4, 8, 12, 16, training both with/without visual traces for each value. We then find the total inference time for each case as the product of the time for

a single prompt and the number of prompts needed per episode. We normalize all three values (total inference time, inference time for single prompt, and number of prompts) relative to the n=16 case, which is what we used in the main paper. It can be seen that the two factors of single prompt inference time and number of prompts needed should be balanced appropriately. For instance, the n = 4 visual traces model has the highest accuracy, but it takes double the time that the n=16 visual traces model does due to higher frequency of prompting. Thus, we conclude that the n=16 steps model is both reasonably efficient and accurate.

# B  Additional Dataset details

Here, we provide more information about our constructed dataset.

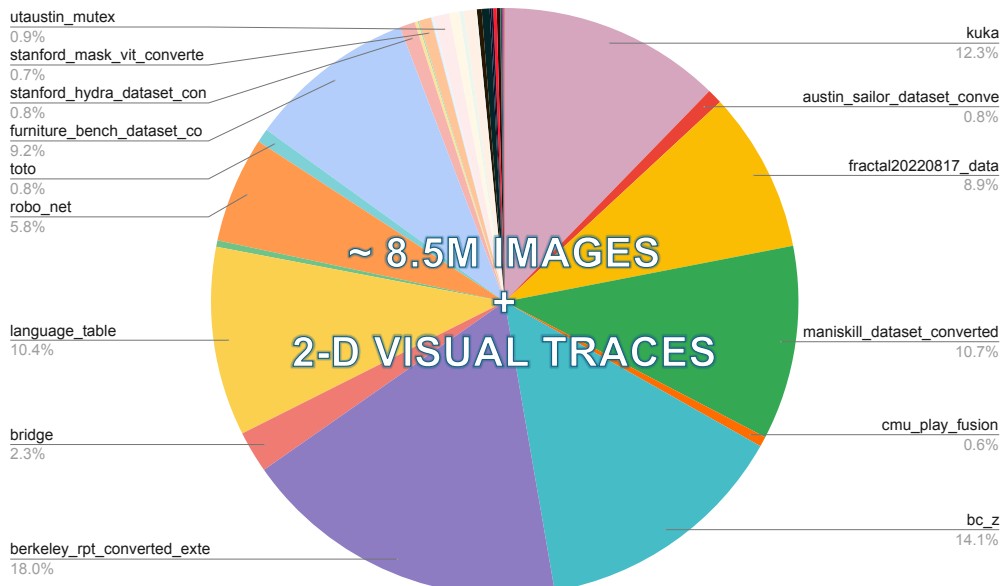

Figure 6: **Data distributions.** We do vision-action instruction pre-training for LLARVA on a dataset built upon Open X-Embodiment [10], including 8.5M image-2-D visual trace pairs.

**Data Distribution**. As mentioned in the paper, we construct the vision-action tuning dataset from a subset of Open X-Embodiment (OXE) [10]. We excluded OXE subsets with poor image quality, smaller image resolution, ambiguous action spaces, or those with widely different robot morphologies, such as Autonomous Mobile Robots (AMRs, which involve locomotion), resulting in 8.5M image-text pairs, whose distribution is shown in Figure 6 and Table 9. Overall, we ensured that the resulting dataset contains subsets of [10] that use end-effector control and joint control, in addition to including both absolute and delta control modes.

**The Generation of 2-D Visual Traces**. The 2-D visual traces can be seen as a trace of the end-effector location in the image plane across time. To generate these traces, we trained an object detector to locate the end-effector from input 2-D images. A bounding box detector was used as opposed to a single point detector because there exist very stable implementations of such models with pre-trained weights publicly available. Specifically, we use the Detectron2 [12] implementation of Faster R-CNN [52] to obtain bounding boxes enclosing the end-effector, and then use the center point of the bounding boxes as the end-effector keypoint. For training, we first loaded the ResNet-101 weights pre-trained on ImageNet. We then randomly collected 200 images from each OXE subset, split the collected 6400 images into a training and validation set with a ratio of 9:1, and trained the detector for 40 epochs using a batch size of 16 and learning rate of $2^{-3}$. Some examples of the resulting detector training set are shown in Figure 7, where the 2-D visual traces are shown in yellow. Note that the traces are a sequence of 2-D coordinates, and Figure 7 is a visualization of

| OXE Subset | Number of Image + 2-D visual trace pairs |
|---|---|
| kuka | 1044466 |
| austin_sailor_dataset_converted_externally_to_rlds | 70758 |
| fractal20220817_data | 753647 |
| maniskill_dataset_converted_externally_to_rlds | 909568 |
| cmu_play_fusion | 47115 |
| bc_z | 1198963 |
| berkeley_rpt_converted_externally_to_rlds_new | 1533451 |
| bridge | 195745 |
| language_table | 885876 |
| stanford_kuka_multimodal_dataset_converted_externally_to_rlds | 30128 |
| robo_net | 496454 |
| toto | 65527 |
| furniture_bench_dataset_converted_externally_to_rlds | 786692 |
| stanford_hydra_dataset_converted_externally_to_rlds | 72160 |
| ucsd_pick_and_place_dataset_converted_externally_to_rlds | 13545 |
| kaist_nonprehensile_converted_externally_to_rlds | 6512 |
| stanford_mask_vit_converted_externally_to_rlds | 57012 |
| utokyo_pr2_opening_fridge_converted_externally_to_rlds | 2276 |
| berkeley_fanuc_manipulation | 11854 |
| utaustin_mutex | 72461 |
| taco_play | 47780 |
| berkeley_autolab_ur5 | 19621 |
| austin_sirius_dataset_converted_externally_to_rlds | 56101 |
| columbia_cairlab_pusht_real | 5486 |
| stanford_robocook_converted_externally_to_rlds | 22894 |
| roboturk | 37120 |
| berkeley_cable_routing | 7797 |
| nyu_franka_play_dataset_converted_externally_to_rlds | 9118 |
| jaco_play | 15515 |
| viola | 15146 |
| tokyo_u_lsmo_converted_externally_to_rlds | 2398 |
| austin_buds_dataset_converted_externally_to_rlds | 6771 |
| dlr_sara_pour_converted_externally_to_rlds | 2695 |
| utokyo_xarm_pick_and_place_converted_externally_to_rlds | 1381 |
| utokyo_pr2_tabletop_manipulation_converted_externally_to_rlds | 6545 |
| dlr_edan_shared_control_converted_externally_to_rlds | 746 |
| dlr_sara_grid_clamp_converted_externally_to_rlds | 1543 |

Table 9: **The vision-action instruction tuning dataset.**

2D Visual Trace

| Subset | Visual Observations | Robot | Control Mode |
|---|---|---|---|
| fractal20220817_data | | Franka | End effector control |
| berkeley_rpt_converted_externally_to_rlds | | Franka | Joint control |
| toto | | Franka | End effector control |
| berkeley_autolab_ur5 | | UR5 | End effector control |

Figure 7: **A few samples from our constructed vision-action tuning dataset.** We visualize some samples of the instruction tuning dataset used in the pre-training stage of LLARVA, with the corresponding robot type and control mode.

these sequences. During training the sequences are predicted in language token space and compared to ground truth.

We further note that the end-effector detector is fairly robust as it has been trained on images that cover a wide range of lighting conditions, cameras, and environments. In addition, it can easily be fine-tuned with a very small number of sample images ( 200) for a new setup, making this approach adaptable.

| Metric | AP | $AP_{50}$ | $AR_{10}$ |
|---|---|---|---|
| Value | 75.2 | 96.3 | 87.2 |

Table 10: Evaluation results for the end-effector detector.

## C  Additional Implementation Details

### C.1  RLBench Experiments

LLARVA is evaluated on 18 tasks from RLBench. All RLBench tasks include two or more variations of a language instruction describing the goal. For example, there might be three variations of the instruction for the same task: *"open the top drawer"*, *"grip the top handle and pull the top drawer open"* and *"slide the top drawer open"*. For simplicity, we use the first instruction variant for training. Below, we describe the RLBench tasks we use for simulator evaluation, along with any modifications we made to the tasks. The intention behind the modifications is to increase the variations of the tasks, such as adding distractor objects with different colors. This exercises the model's language grounding abilities. All tasks are unmodified unless otherwise noted.

**Training Setup**. We start with a LLARVA model that has undergone vision-action instruction pre-training on OXE as described in Section 2.3, and perform step 2 (Section 2.3) instruction fine-tuning for four epochs on task-specific downstream data (e.g., picking, stacking, destacking) using eight A6000 GPUs. Step 2 instruction tuning is done using 800 demonstrations for each RLBench task. The domain gap between step 1 and step 2 is large as we change from almost entirely real data to simulation while at the same time changing robots and tasks. We note that while other works train on a smaller amount of data, they use roughly the same order of magnitude of data as LLARVA, and exploit the power of 3-D representations. For example, PerAct [7] uses 100 examples per task but exploits voxel-based 3-D representations, which are rare and difficult to obtain. Our approach has

the advantage of being able to leverage 2-D representations, which may require additional data but with roughly the same order of magnitude as methods that utilize 3-D.

**Open Drawer**. The task is to open one of three drawers. The success metric is a full extension of the prismatic joint of the target drawer.

**Meat off Grill**. The task is to take either a piece of chicken or steak off the grill and put it on the side. The success metric is the placement of the specified meat on the side, away from the grill.

**Turn Tap**. The task is to turn either the left or right handle of the tap. Left and right are defined according to the orientation of the faucet. The success metric is the joint of the specified handle being at least $90°$ away from the starting position.

**Put Money**. The task is to pick up the stack of money and place it on the specified shelf of a safe. The safe has three shelves: top, middle, and bottom. The success metric is the placement of the stack of money on the specified shelf in the safe.

**Push Buttons**. The task is to push the colored buttons in the specified sequence. There are always three buttons present in the scene, whose colors are sampled from 20 options, and the number of buttons to press is between one and three. The success metric is all specified buttons being pressed in the right order.

**Sweep Dustpan**. The task is to sweep the dirt particles into the specified dustpan. There are two dustpans, one short and one tall, and both are always present in the scene. The success metric is all five dirt particles being inside the specified dustpan. We modified this task by adding a variation with a different-sized dustpan.

**Slide Block**. In this task there is a block and four colored squares in the scene (green, blue, pink, and yellow). The task is to slide the block onto either the green or pink squares. The success metric used is some part of the block being on the specified target square. The original task only had one target square, and we modified it by adding three additional colored squares — one target and two distractors.

**Close Jar**. The task is to screw in the lid on the jar with the specified color. There are always two colored jars in the scene, one target jar and one distractor jar. The success metric used is the lid being on top of the specified jar and the robot gripper not grasping any object. We modified this task so that the target jar color is drawn from a list of two possible colors (blue or teal). The color for the distractor jar was still chosen out of 20 options.

**Screw Bulb**. There are two bulb holders of different colors, and the task is to pick up a light bulb from the stand specified by color and screw it into the bulb stand. The color of the target holder is sampled from two colors, while the color of the distractor holder is sampled from the original 20 color options. The success metric used is the bulb from the specified holder being inside the bulb stand. We modified this task to use two colors for the target holder (yellow and purple) rather than 20 as in the original task specification.

**Place Wine**. The task is to pick up the wine bottle and place it at the specified location in a wooden rack. The rack has three locations: left, middle, and right. The success metric is the placement of the bottle on the specified location in the rack.

**Reach and Drag**. The environment has a cube, a stick, and four possible colored target squares. The task is to pick up the stick and use it to drag the cube to the target square of a specified color. The other three squares are considered distractors. The success metric used is some part of the block being inside the target's area. We modified this task to sample the target color from a list of three colors (maroon, magenta, teal). The colors for distractor squares are still sampled from 20 options.

**Stack Blocks**. The scene starts with 8 blocks and a green platform. Four of the blocks are of a target color, and the other four have a distractor color. The task is to stack $N$ blocks of the target color on the green platform. The success metric is $N$ blocks being inside the area of the green platform.

**Put Item in Drawer**. There is a block kept on top of a chest of closed drawers. The task is to place the block into the specified drawer among three possible options: top, middle, or bottom. The success metric is the placement of the block inside the specified drawer.

**Sort Shape**. The scene has four distractor shapes and one correct shape. The task is to pick up the shape specified in the language instruction and place it in the correct hole in the sorter. The success metric is the correct shape being inside the corresponding hole.

**Insert Onto Square Peg**. The scene has a platform with three differently colored pegs, and one square shaped object with a hole in the middle. The three colors are sampled from 20 color instances. The task is to pick up the square and put it on the peg specified in the language instruction, with the success metric being the placement of the square fully on the peg.

**Stack Cups**. The scene has three cups with colors sampled from 20 options. The task is to stack all cups inside the cup specified in the language instruction. The success metric for this task is all other cups being inside the specified cup.

**Put Groceries in Cupboard**. The scene always has nine grocery items and one cupboard. The task is to place the item specified in the language instruction inside the cupboard. The success metric used is the placement of the item inside the cupboard.

**Place Cups**. The scene always has one cup holder with three spokes and three cups with handles. The task is to place $N$ of the cups on the cup holder ($N \in \{1, 2, 3\}$). The success metric used is the alignment of each cup's handle with a spoke on the cup task.

**Toilet Seat Down**. The scene consists of a toilet which initially has its seat up. The task is to put the toilet seat down. The success metric used is the joint of the toilet seat being at an angle consistent with the seat being fully down.

**Close Laptop Lid**. The scene consists of a laptop which initially has its lid open. The task is to close the laptop. The success metric used is the joint of the laptop lid being at an angle such that the screen is fully down.

**Put Knife on Chopping Board**. The scene consists of a knife inside a knife holder, and a chopping board. The task is to pick up the knife from the holder, and place it on the chopping board. The success metric used is the knife being on the surface of the chopping board, and the robot gripper not grasping anything.

**Put Umbrella in Umbrella Stand**. The scene consists of an umbrella and an umbrella holder. The task is to pick up the umbrella and put it into the stand. The success metric used is the umbrella being inside the stand, and the robot gripper not grasping anything.

**Move Hanger**. The scene consists of a clothes hanger and two racks. The task is to move the hanger from its current rack to the other rack. The success metric used is the hanger being placed on the other rack.

## C.2 Real Robots Experiments

**Hardware Setup**. We use a Franka Emika Panda robot with a Franka gripper for real robot data collection and evaluations. A Logitech BRIO 4K camera positioned to the right of the Franka robot provides single-view RGB (without depth data) vision input to our model, as shown in Figure 8. Camera autofocus is disabled, and the data is captured at 640x480 resolution. The model inference is done on a 48GB NVIDIA A6000.

**Data Collection**. We use the data collection code and process from https://github.com/Max-Fu/franka-scripted to collect data for picking, stacking, and destacking tasks. The script generates data for an arbitrary number of episodes. For each episode, the process generates x-y positions on the table plane using a uniform random distribution for each axis. The script directs the robot to place the cube at each location and then collects the camera and joint information as the robot is

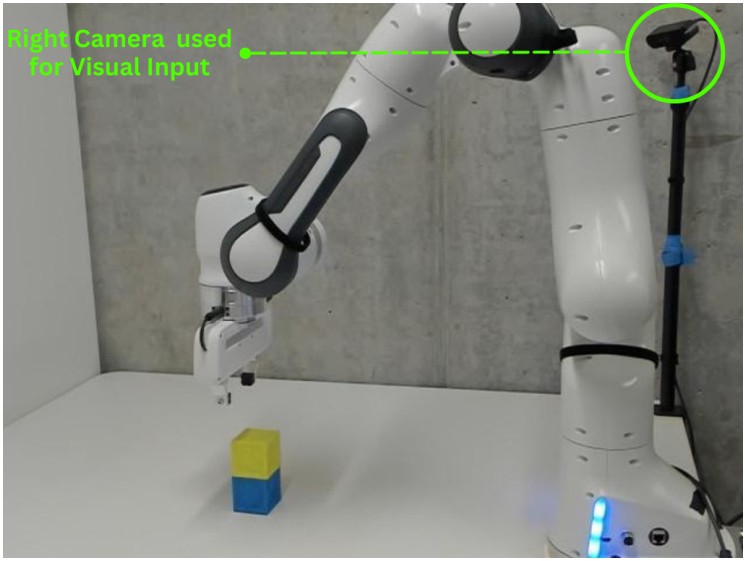

Figure 8: The real robot setup with Franka Emika Panda used for evaluating LLARVA.

directed to pick, stack, or destack the cubes. Vision is not used during this process as the cube locations are all generated and therefore known.

**Training and Execution**. For the Franka Emika Panda robot experiments, we start with our LLARVA model that has undergone vision-action instruction pre-training on OXE as described in Section 2.3, and perform step 2 (Section 2.3) instruction fine-tuning for four epochs on 1920 episodes of task-specific downstream data (e.g., picking, stacking, destacking) using 8 A100 GPUs. This is similar to other baselines, such as RPT [16], that uses an equal amount of in-domain episodes (1920) for pre-training, with an additional 120-240 episodes used for fine-tuning depending on the task. Additionally, [16] uses three camera views for each episode, while LLARVA uses only one. Nevertheless, it can be observed that LLARVA demonstrates superior performance on all three tasks tested despite using comparable or even fewer episodes. Finally, each real robot evaluation consists of 16 repeated pick, stack, or destack operations at a random x-y location on the table plane for each repetition. We report the success rate of the 16 operations.

## D   Discussion, Limitations and Future Work

While LLARVA offers substantial benefits for enhancing robot learning across various environments, it is important to recognize certain limitations that accompany our approach. First, even though LLARVA already shows generalization capabilities using our instruction language format, we think future instructions should include 3-D information about the real world. Secondly, it is necessary for LMMs to be able to process multiple views leveraging depth or voxels, which may require the use of interleaved tokens as is done in a few existing LMMs. We therefore believe that the next robotic LMMs should utilize this technology, and we leave this for future work. This presents a great opportunity for future work on next-generation instruction-tuned LMMs for robotics.

## E   Licenses and Privacy

The license, PII, and consent details of each dataset are in the respective papers. In addition, we wish to emphasize that the datasets we use do not contain any harmful or offensive content, as many other papers in the field also use them. Thus, we do not anticipate a specific negative impact, but, as with any machine learning method, we recommend exercising caution.

