# OpenReview forum: "LLARVA: Vision-Action Instruction Tuning Enhances Robot Learning"
_robot-learning.org/CoRL/2024/Conference — CoRL 2024_

### Official Review · Reviewer_WWDV · 2024-07-21
**The paper introduces LLARVA, a Large Multimodal Model (LMM) model instruction-tuned on structured language prompts unifying diverse robotic configurations and visual representations showing the trace of the robot's end effector.**

**Originality:** 2
**Technical Quality:** 3
**Clarity Of Presentation:** 2
**Potential Impact:** 3
**Recommendation:** 3
**Confidence:** 5

**Review:**

The paper presents LLARVA, an instruction-tuned Large Multimodal Model (LMM) for robotics. The proposed method of using structured prompts and visual traces is well-conceived and effectively addresses the challenges of diverse robotic configurations and aligning vision and action spaces, respectively.

Strengths:

1. The use of 2D visual traces to align vision and action spaces is simple and effective, and addresses an important problem in existing robotic LMMs.
2. The model is evaluated on a wide range of tasks both in simulated and real-world environments, demonstrating its robustness and versatility.
3. LLARVA outperforms all baselines with 2D visual representations without visual traces on most tasks in RLBench, particularly in generalization across different robot configurations and environments.

Weaknesses:

1. While the use of 2D visual traces in LMMs is underexplored, visualizing the robot's trajectory has been studied in numerous previous work [1, 2, 3, 4, 5] and limits the novelty of the proposed approach.
2. Is this end-effector detector robust on camera/lighting/environment change? Is this method scalable, or is it only applicable in in-distribution data? If not, how could these issues be resolved in the future?
3. Especially in case of fixed camera viewpoint, you can easily know camera intrinsics and extrinsics, and get (perfect in simulation environment) the projection mapping from the joint positions to the exact 2D visual trace. What is the advantage of using a trained detector in fixed camera viewpoint setting, despite the discrepancy between ground truth and predicted visual traces as shown in Figure 3?
4. Why does LLARVA predict both visual trace and next action? It seems the paper assumes an exocentric fixed camera viewpoint (please correct if wrong), there exists an exact projection function from a 3D end-effector trajectory to its corresponding 2D visual trace in the camera observation. Since LLARVA is predicting both visual trace and next action, they might not match since there is no hard constraint while training to ensure that the projection is correct.
5. L257: Object-centric Grounding for Vision Tasks in the related work section seems less relevant to the paper. I recommend having a section for visual motion representations, especially on visualizing the end effector trajectory.

**Quality Of The Limitations Section:**

1

**Questions For Rebuttal:**

1. L98: It seems the task horizon is fixed for each training. How does the proposed approach become robust and adaptable across a spectrum of task durations?
2. How is the bounding box end effector detector trained? What is the reason the detector directly outputs a bounding box, instead of a single point? What is the loss function? Were there any concerns using the trained detector, since it is only trained on 200 samples per robot?
3. Extending the paper's setup into a dynamic camera viewpoint, how can the proposed method be adopted to the new setup?
4. How were the RLBench data collected and labeled? Based on Section 3.2, I assume that there were both successful and failed episodes. Please share the data generation and success labeling procedure more in detail.
5. Can the model be easily adapted to incorporate 3-D visual traces or other forms of 3-D data?
6. Why do Image-BC baselines perform so poorly, scoring 1.3% success rate? What is the performance of the proposed model without visual traces in RLBench?
7. What is the performance of the pre-trained model on Sawyer on the right table of Table 3? Could the authors clarify if the 'Franka' row is the performance of the pre-trained model, or the model fine-tuned on a small Franka/Sawyer robot dataset?
8. In the limitations section, why do the authors think that future instructions should necessarily include 3D information and multiple views? This seems to contradict the results, since the results in the paper show that using the proposed visual representation outperforms using 3D representation or multiview observations.

[1]  J. Gu, S. Kirmani, P. Wohlhart, Y. Lu, M. G. Arenas, K. Rao, W. Yu, C. Fu, K. Gopalakrishnan, Z. Xu, et al. Rt-trajectory: Robotic task generalization via hindsight trajectory sketches. arXiv preprint arXiv:2311.01977, 2023.

[2] H. Bharadhwaj, R. Mottaghi, A. Gupta, and S. Tulsiani. Track2act: Predicting point tracks from internet videos enables diverse zero-shot robot manipulation. arXiv preprint arXiv:2405.01527, 2024.

[3] S. Nasiriany, F. Xia, W. Yu, T. Xiao, J. Liang, I. Dasgupta, A. Xie, D. Driess, A. Wahid, Z. Xu, et al. Pivot: Iterative visual prompting elicits actionable knowledge for vlms. arXiv preprint arXiv:2402.07872, 2024.

[4] M. Vecerik, C. Doersch, Y. Yang, T. Davchev, Y. Aytar, G. Zhou, R. Hadsell, L. Agapito, and J. Scholz. Robotap: Tracking arbitrary points for few-shot visual imitation. arXiv preprint arXiv:2308.15975, 2023.

[5] C. Wen, X. Lin, J. So, K. Chen, Q. Dou, Y. Gao, and P. Abbeel. Any-point trajectory modeling for policy learning. arXiv preprint arXiv:2401.00025, 2023.

**Robotics Focus:**

4

**Summary Of Paper:**

LLARVA (Large LAnguage model for Robotic Vision and Action) is a Large Multimodal Model (LMM) instruction-tuned on structured language prompts unifying diverse robotic configurations. LLARVA also uses 2D visual representation visualizing the trace of the robot's end effector. to align vision and action spaces effectively. Pre-trained on 8.5 million image-visual trace pairs from the Open X-Embodiment dataset, LLARVA is evaluated on 12 tasks in the RLBench simulator and a physical Franka Emika Panda robot in the real-world. The results indicate that LLARVA outperforms several baselines, demonstrating strong performance and generalization across robots.

**Summary Of Recommendation:**

The paper integrates visual trace and structured language prompts to make a robust and generalizable LMM for robots. However, the above comments reduce the clarity and the overall assessment of the work. I would like to see the authors' response on the weaknesses and questions mentioned above.

---

### Official Review · Reviewer_Uugi · 2024-07-21
**LLARVA: Vision-Action Instruction Tuning Enhances Robot Learning**

**Originality:** 4
**Technical Quality:** 3
**Clarity Of Presentation:** 4
**Potential Impact:** 3
**Recommendation:** 3
**Confidence:** 5

**Review:**

In this study, the authors present LLARVA, an open-source instruction-tuned LMM to predict robotic actions for different tasks, environments, controls, and robots using given visual and language representations.

The idea and the motivation are very clear. The paper is well-written and easy to understand. You can find my comments below:

It would be beneficial to see an ablation study for only-vision and only-language modalities, as both are presented as crucial for a detailed understanding of the environment, task, and robotic actions.

Fine-tuning and its limitations are not clear in the paper. The authors should explain what needs to be done for a new task/environment/robot if one wants to use this method off the shelf. I understand that fine-tuning improves accuracy as it introduces the new configuration to the model as training data with ground truths. However, it is only presented as correcting real-world properties. Isn't OXE already a robotic dataset based on physical real-world properties? This should be discussed further.

The authors present their method as capable of recovering from failures. I wonder what would happen when multiple failures occur such that t approaches an average episode length for the given task. For example, let's assume a stacking action usually takes 60 steps; what happens when t becomes 120 and the robot has still not grasped the object due to prolonged failures? Does it prioritize reaching the final position, or can it still prioritize grasping the object? It is also important to state whether h is ever reset to zero after a long time or failure.

It would also be really powerful if there was an experiment showing that the method is accurate for a new robot that was not in the training set. It would also support the fact that fine-tuning is an important aspect of this method since these are new test-time additions and modifications.

My biggest concern is that, despite being convinced this is a really powerful framework, I still think the current evaluations and, most importantly, interpretations of the results are not sufficient to demonstrate it. There is a significant lack of interpretation of the results and why they are good, bad, or important. These should be further explained after stating that the method outperforms the benchmarks. Without the authors' insights, any aspect of the paper, whether it is really this amazing or not, is just a wild guess. For example, the accuracy difference in the ablation study for using visual traces drastically changes depending on the task. Why is that happening? Is it because the visual traces only track the end-effector but not the joint space so that the large improvements are for end-effector control tasks and the small improvements are for joint control tasks? This is a wild guess for a reader that the authors should have insight into.

Minor edits:

Line 181: change "strong" to "stronger."

Line 188: specify exactly how many demonstrations when mentioning "a comparable number of demonstrations."

Equation 3 states A and P are ground truths but uses the ^ notation, confusing the reader.

**Quality Of The Limitations Section:**

2

**Questions For Rebuttal:**

Why do the ablation study accuracies for the visual traces differ this much according to the task? Is it related to task difficulty, robot, controller type, etc.?

What happens when you use a new robot, task, controller, or environment? Please answer in the light of pre-training and fine-tuning, with examples from experiments if possible.

Would using depth or RGBD images eliminate the problem of 2D to 3D robot action mapping? Does the dataset have these types of data?

Since it is shown as an upper bound in the experiments, what are the advantages of using this method over other accurate methods shown in the paper using 3D representations? Please further explain by comparing possible aspects.

**Robotics Focus:**

4

**Summary Of Paper:**

In this study, the authors present LLARVA, an open-source instruction-tuned LMM to predict robotic actions for different tasks, environments, controls, and robots using given visual and language representations. They pre-trained an LMM using 8.5M data and evaluated it on 12 RLBench tasks and real-world, showing it out-performs similar benchmarks.

**Summary Of Recommendation:**

I am convinced this is a really powerful framework with clear novelties and improvements, but I still think the current evaluations and, most importantly, interpretations of the results are not sufficient to demonstrate it.

---

### Official Review · Reviewer_s7WX · 2024-07-22
**Reviewer for LLARAVA**

**Originality:** 3
**Technical Quality:** 3
**Clarity Of Presentation:** 3
**Potential Impact:** 3
**Recommendation:** 3
**Confidence:** 5

**Review:**

**Strengths:**

1. The paper models the action prediction task as natural language generation task by formating the robot’s input in language form and predicting the actions token by token which propose a new way to employ off-the-shelf LMM into robotics.
2. The paper finetuned a LMM based on LLaVA which using LoRA, and they use ViT as vision backbone while Llama-7B as language decoder. LLaVA is totally open-sourced making it easy to follow.
3. The authors generates 8.5M image-text visual trace pairs from the Open-X Embodiedment dataset to pretrain the LMM which has 13 different types of robots. Then finetuning the model with downstream tasks. They showed that pretraining helps model adapt to specific downstream settings.
4. The authors report the results both in simulated and real scenarios.
5. The authors propose to predict 2D positions of the end effector first instead of directly predicting the action(like joint position) which improves the performance.

**Weaknesses:**

1. All 3 tasks in real scenarios are pick&place type which are too simple to complete. Pick cube only needs lift up the cube which only consists of the variation of z-axis. And only a change occurred in the y-axis for the visual traces. Stack and destack cube are mostly identical as well as the visual traces except the height of pick and place. It’s hard to tell whether the model can predict more complicated traces in reality.
2. In abstract, the authors said they introduce LLARVA to address the generalization in different settings. But the paper only validate this generalization in RLBench with four tasks across two kinds of robotic arm. Both sawyer and franka are 7 Dof robotic arms which means the difference is tiny. And the model is separately finetuned with the data corresponding to the setting. As a result, it’s not sufficient to demonstrate the model’s generalizability.
3. In section 3.1, the authors claimed that they utilized the same language encoder, ViT and projection layer in LLaVA-1.5. And LLaVA-1.5 has two model sizes: 7B and 13B. Both sizes of model are computational heavy during real-time robot control.  As seen in figure 1, the model need to predict all the remaining visual traces which largely increase the numbers of tokens. Though the authors mentioned that the model can predict next n actions, they didn’t report a comprehensive  comparison of task success rates when increasing n.
4. In the video, the ‘sweep to dustpan’ part, the variation of predicted visual traces is significant and deviates greatly from the ground truth. And l concern if visual traces is wrong, does this really help improve the accuracy of predicted actions? And the table 3 left only involves 4 tasks which l think not enough to validate the effectiveness of predicting 2-D visual trace.

**Quality Of The Limitations Section:**

2

**Questions For Rebuttal:**

1. Experimental setup in section 3.3, the evaluate strategy is not clear, please explain why the numbers of evaluation episodes are different for each method.
2. It will be beneficial to provide more experiments on inference efficiency(especially the number of predicted steps) and more different robot settings mentioned in weaknesses.
3. It would be helpful to provide more experiments on the ablation of the 2-D visual trace.
4. In line 152, it said there is a detector trained specifically on OXE and l am curious about how to get original training data and data quantity.
5. The OXE dataset includes various types of robotic arms with different degrees of freedom and shapes. I would like to ask if the robotic arms selected from the dataset all have the same degrees of freedom？

**Robotics Focus:**

4

**Summary Of Paper:**

The paper transfers the robot action prediction task into natural language generation task through instruction tuning. The paper generate 8.5M image-visual trace pairs from the Open X-Embodiment dataset as pretraining dataset.  They show that pretraining improves performance over non-pretraining. They propose predicting 2D visual traces before the next step action, which helps improve performance as well. They finetuned a LMM based on LLaVA by LoRA.

**Summary Of Recommendation:**

In total, this work is promising and provides new angle of solving robot action prediction problem by transfer it into natural language generation problem. The proposed method is technically sound. However, this work necessiates more experiments to validate the model’s ability to generalize in different settings. What’s more, tasks in real scenarios are less and simple.

---

### Author Rebuttal · Authors · 2024-08-09

Please find attached our video demonstrations for:

- **New task experiment (push buttons):** Two demo videos, "NEW_TASK_three_buttons_viz.mp4" and "NEW_TASK_one_button_viz.mp4", that each show a successful completion of these tasks along with a visualization of the predicted visual traces.

- **New robot experiment (UR3):** Two demo videos, "NEW_ROBOT_stack_ur3_viz.mp4" and "NEW_ROBOT_pick_ur3_viz.mp4", that each show a successful completion of these tasks on a 6 DoF UR3 robot with a 6 DoF five-fingered end-effector in a new setup. The videos also have a visualization of the predicted visual traces.

---

### Decision · Program_Chairs · 2024-09-04

**Decision:**

Accept

**Comment:**

The reviewers appreciate the  presented approach of modeling the action prediction task as a language generation task. The idea is well-motivated and clearly communicated, and the approach was evaluated on both simulated and real-world experiments. However, the reviewers have identified several shortcomings that should be addressed before publication:

- All three real-world tasks are basic pick-and-place tasks that don’t require complicated trajectories/traces to succeed, leaving doubt about the real-world capability of the approach.
- The paper claims generalization in different settings, but the validation is only on two robots in RLBench.
- Vision-only and language-only ablations would strengthen the paper further.
- The paper should elaborate on the fine-tuning procedure for a new task/environment/platform.
- The experiment/results should be discussed in more detail.
- Additional clarification on why an exocentric fixed camera perspective does not lead to a mismatch between the end-effector trajectory and visual trace would be beneficial.

## Post Rebuttal
The rebuttal addressed all major concerns of the reviewers and was well received. The reviewers unanimously agree that this work should be presented at CoRL 2024. However, some clarifications and details should still be incorporated into the main manuscript as agreed upon by the authors.